# "SDM:HOSP"- a generic model for hospital-based implementation of shared decision making

Karina Dahl Steffensen[1,2], Dorte Gilså Hansen[1,2], Kurt Espersen[3], Susanne Lauth[4,5], Peter Fosgrau[6], Anders Meinert Pedersen[7], Peter Sigerseth Groen[8], Christian Sauvr[8], Karina Olling [1] *

1 Center for Shared Decision Making, Department of Clinical Oncology, Lillebaelt University Hospital of Southern Denmark, Vejle, Denmark, 2 Institute of Regional Health Research, Faculty of Health Sciences, University of Southern Denmark, Odense, Denmark, 3 Region of Southern Denmark, Vejle, Denmark, 4 West Jutland Hospital of Southern Denmark, Esbjerg, Denmark, 5 Aarhus University Hospital, Aarhus, Denmark, 6 Hospital of Southern Denmark, Aabenraa, Denmark, 7 Mental Health Services, Vejle, Region of Southern Denmark, 8 Department of Clinical Development, Odense University Hospital, Odense, Denmark

* Karina.olling@rsyd.dk

**Data Availability Statement:** All relevant data are within the paper and its Supporting Information files.

## Abstract

### Background

Shared decision making (SDM) is a core element in the meeting between patient and health-care professionals, but has proved difficult to implement and sustain in routine clinical practice. One of five Danish regions set out to succeed and to develop a model that ensures lasting SDM based on learnings from large-scale real-world implementation initiatives that go beyond the 'barriers' and 'facilitators' research approach. This paper describes this process and development of a generic implementation model, SDM:HOSP.

### Methods

This project was carried out in the Region of Southern Denmark with five major hospital units. Based on existing theory of SDM, SDM implementation, implementation science and improvement methodology, a process of four phases were described; development of conceptual elements, field-testing, evaluation, and development of the final implementation model. The conceptual elements developed aimed to prepare leaders, train SDM teachers, teach clinicians to perform SDM, support development of patient decision aids, and support systematic planning, execution and follow-up. Field testing was done including continuous participant evaluations and an overall evaluation after one year.

### Results

Data from field testing and learnings from the implementation process, illustrated the need for a dynamic and easy adjustable model. The final SDM:HOSP model included four themes; i)Training of Leaders, ii) Training of Teachers and Clinicians, iii) Decision Helper, and iv) 'Process', each with details in three levels, 1) shared elements, 2) recommendations, and 3) local adaption.

**Funding:** The authors received no specific funding for this work.

**Competing interests:** The authors have declared that no competing interests exist

## Conclusions

A feasible and acceptable model for implementation of SDM across hospitals and departments that accounts for different organizations and cultures was developed. The overall design can easily be adapted to other organizations and can be adjusted to fit the specific organization and culture. The results from the ongoing and overall evaluation suggest promising avenues for future work in further testing and research of the usability of the model.

## Introduction

The contact between patients and the healthcare system is characterized by decisions. Aside from the more acute contacts where it may be time challenging, all significant healthcare decisions should be based on a dialogue that incorporates both the healthcare professional's experience and expert knowledge of illness and health, as well as the patient's experience, values and preferences for their own lives. This overarching premise should apply in all situations where a healthcare professional meets a patient.

Shared Decision Making (SDM) is an approach that describes a simple fact: Patients are no longer just recipients in healthcare, and healthcare professionals are not transmitters. The patient is the one for whom the treatment has the greatest consequence and a paternalistic approach is no longer perceived adequate. SDM is an essential element of patient engagement in own treatment and care, a new way of thinking in healthcare, which is about involving the patient more in treatment [1, 2]. It is an approach that leads to better-informed decisions that to a greater extent align with what matters most to patients [3]. In doing so, SDM not only contributes to better patient outcomes but also supports a growing recognition that patient involvement is a moral obligation.

SDM can be defined as a collaboration between the patient and the healthcare professional when decisions have to be made about screening, diagnostics, treatment, care and follow-up to the extent and in the ways that the patient wants. SDM implies that the patient receives the best available information about the possible different options, their advantages, disadvantages and uncertainties, and is provided support to identifying his or her values and preferences and in choosing the option that best matches his or her preferences. SDM is a patient-centered approach that asks the question "what matters most to you when thinking about the different choices."

Implementing new cultures in healthcare is difficult. Despite good intentions and policy statement describing better and broader implementation of shared decision making principles into clinical routine practice, this has not occurred [4–7]. Recognition of cultural and practical barriers have partly explained this difficulty, e.g. lack of accessible knowledge, skills and experience about SDM methods as well as lack of adaption to clinical systems and workflows, missed opportunities to activate and engage patients and insufficient strategies for implementation [8–11]. Moreover, research projects rarely change clinical practice if there is no systematic effort and management of how to implement and spread the efforts that took place in the research project. Therefore, the development of generic models that systematically describe the necessary parts of a successful SDM implementation process is needed. Models that can form the basis for lasting change of future initiatives and that can help clinical teams, hospitals and other healthcare organizations to better understand what implementation of SDM might look like outside of a research setting.

To move from classic delimited research projects to real-world implementation, it can be beneficial to use techniques from implementation science. Available literature on

implementing evidence-based shared decision making tools points to a number of challenges and as described by a paper by Elwyn et al a broader conceptualization and measurement of shared decision-making would provide a more substantive evidence base to guide implementation [12]. Nevertheless, current literature on SDM implementation is itself challenged by the fact that these papers are often based on research data and rarely on knowledge gathered as part of real-world and more widespread implementation outside of a research project [12–14]. In this paper, we describe the implementation of SDM in a large real-world setting across multiple hospitals and within multiple clinical specialties. One of the five Danish regions responsible for public health care has set out to succeed and to develop a model that can ensure lasting SDM based on learnings from large-scale real-world implementation initiatives that go beyond the 'barriers' and 'facilitators' research approach [14]. It is this process and the results in the form of a generic model that are described in this paper.

The aim of the real world large-scale and wide-spread regional implementation project was therefore to implement SDM across hospital sites, medical specialties and clinical departments, evaluate the process and results, and make recommendations on how to implement SDM in real-world settings.

This paper aims 1) to describe the development of a generic model for systematic implementation of SDM into routine clinical practice at somatic and mental health care hospitals and 2) to present the final model SDM:HOSP.

## Methods

This quality improvement project was conducted with a pragmatic approach well supported by research-based expertise, to combine the best from two worlds.

The aim was to create a strong concept with thoroughly tested elements, a strong organization, and a possibility to work with and learn from the clinical real-life improvement work, to establish a mutual frame and model for successful long-term change in a health care organization.

The final model, SDM:HOSP was developed based on evidence-based knowledge and real-life learning (field testing) including ongoing evaluation and revision of the different elements.

The paper is written using the Standards for QUality Improvement Reporting Excellence (SQUIRE guideline 2.0) [15]. Please see S1 File for Squire Checklist.

### Context

Denmark is a relatively small country (5.6 Mio. citizens) characterized by a tax-based health care system. The hospital-based health care is organized in five regions making room for regional adjustment of services within the financial and national regulatory framework. This project was carried out in the Region of Southern Denmark (1.2 Mio. citizens) with five major hospital units; one covering mental health services (PSY) and four covering the somatic health care, Odense University Hospital (OUH) Hospital of Southern Denmark (SHS), West Jutland Hospital of Southern Denmark (SVS) and Lillebaelt University Hospitals of Southern Denmark(SLB) (Fig 1).

The hospitals in the Region of Southern Denmark have 25,000 employees, mainly physicians, nurses, therapists, midwives and other health care professionals.

Several contextual elements were considered important for the project. Regarding implementation of patient involvement in individual health care, the program leaders of Lillebaelt Hospital University Hospital of Southern Denmark were pioneers when pointing out a "patient first" philosophy in 2011 [16]. As another important step, a Center for Shared Decision Making (CFFB) was launched in 2014 [17] at one of the hospitals (Lillebaelt Hospital),

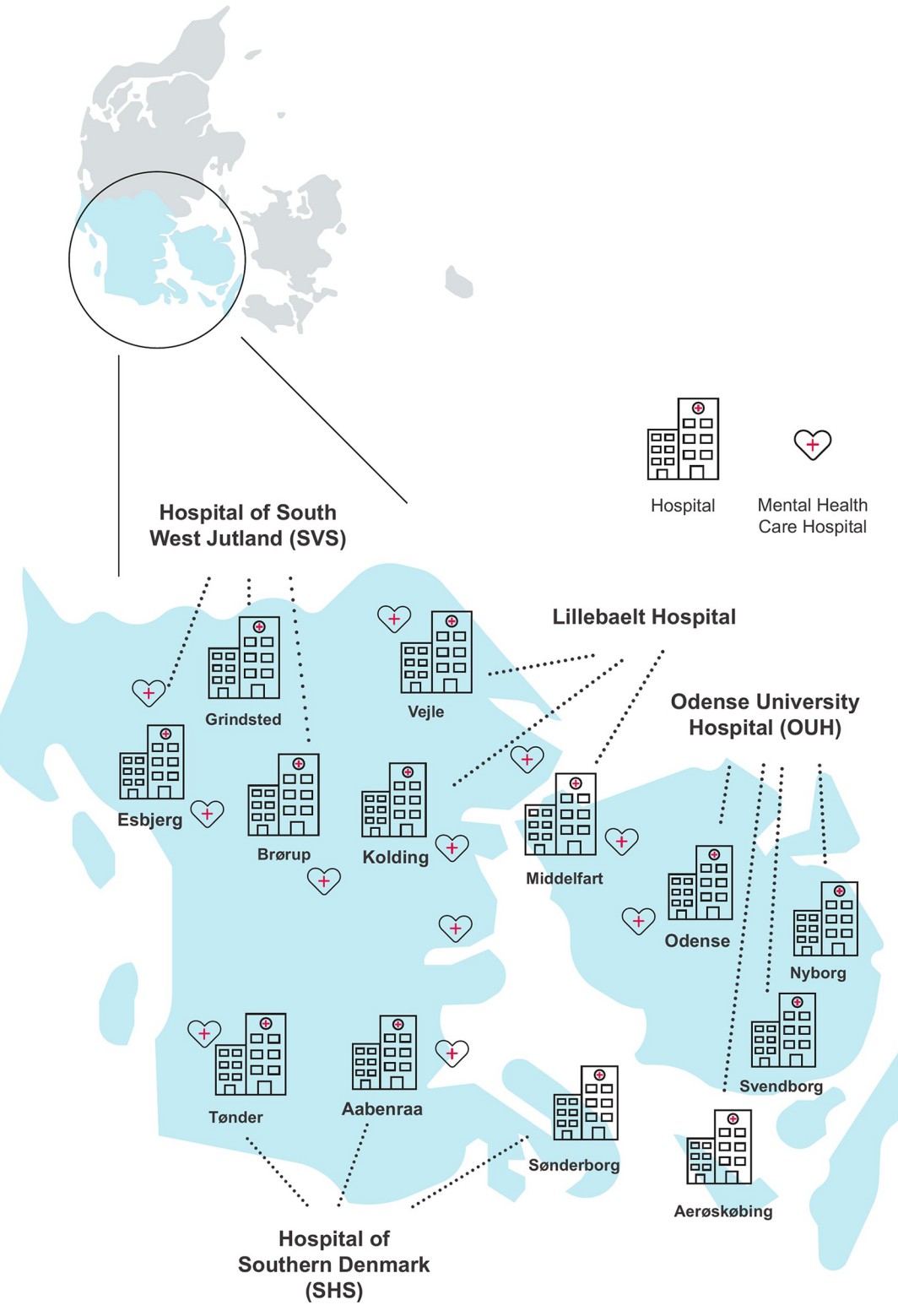

**Fig 1. Hospital units in the Region of Southern Denmark.**

primarily as a research center focusing on SDM, Patient Decision Aids (PtDAs) and patient-reported outcomes.

In the Region of Southern Denmark, there has been a basic focus on working with improvements. There has been developed a "Lean for Leaders" course that all managers in the Region of Southern Denmark complete, to be able to support the improvement work and the change it causes. This "lean for leaders" concept is called "The South Danish Improvement Model" and is highly inspired by cooperation between Virginia Mason Medical Center in Seattle, USA and the Region of Southern Denmark [18].

In line with national health care plans [19, 20] and convincing results from several regional research and demonstrations projects [6, 21–25] it was decided by the regional council of the Region of Southern Denmark to implement SDM at all five hospital units from 2018/2019. This regional implementation initiative was given high priority. A steering committee was granted the overall responsibility for the process. Beyond a member from the executive board of the Region of Southern Denmark and executives from all regional hospital units, the committee included a representative from the Patient and Relatives Council at each hospital unit and the CEO and COO of CFFB (Fig 2).

To improve the process of implementation, each hospital recruited a SDM consultant to support local implementation. CFFB was declared a center of regional center of shared decision making and upgraded with three employees.

## Theoretical framework

The policies, theory and research of SDM call for pragmatic real-life solutions to implement SDM successfully in today's health care, and this project took outset in existing theory of SDM and SDM implementation, implementation science including the System of Profound Knowledge and The South Danish Improvement Model.

W. Edward Deming's System of Profound Knowledge (SoPK) is defined as "the interplay of the theories of systems, variations, knowledge, and psychology" [26]. In 2020 Waldron et.al described a set of key mechanisms detected through a realist synthesis and described a program theory of SDM, which could assist health care professionals, policymakers and patients when practicing and implementing SDM [27]. These key mechanisms and earlier described barriers of implementing SDM [9–11] would all be affected by using the principles of SoPK when leading improvement. When using this system, it is also recognized that there is a relationship between improvement and change and that improvement comes from the action when developing, testing and implementing changes. When understanding the underlying premise of "every system is perfectly designed to deliver the results it produces" [26], understanding the variation, build knowledge and recognize the human side of change, a pragmatic and assessable way of implementing SDM would emerge.

## Interventions

The elements towards a final model were conceptualized and described within a process of four phases: development of conceptual elements, field-testing, evaluation, and development of the final implementation model (Fig 3).

**Phase 1: Development of conceptual elements.** Phase 1 was the development of the conceptual elements, which were developed in cooperation between CFFB and the Regional Steering Committee. These elements aimed to train leaders in their roles, teach clinicians to be SDM teachers, clinicians to perform SDM, support development of PtDAs, and support of systematic planning, execution and follow-up of the implementation

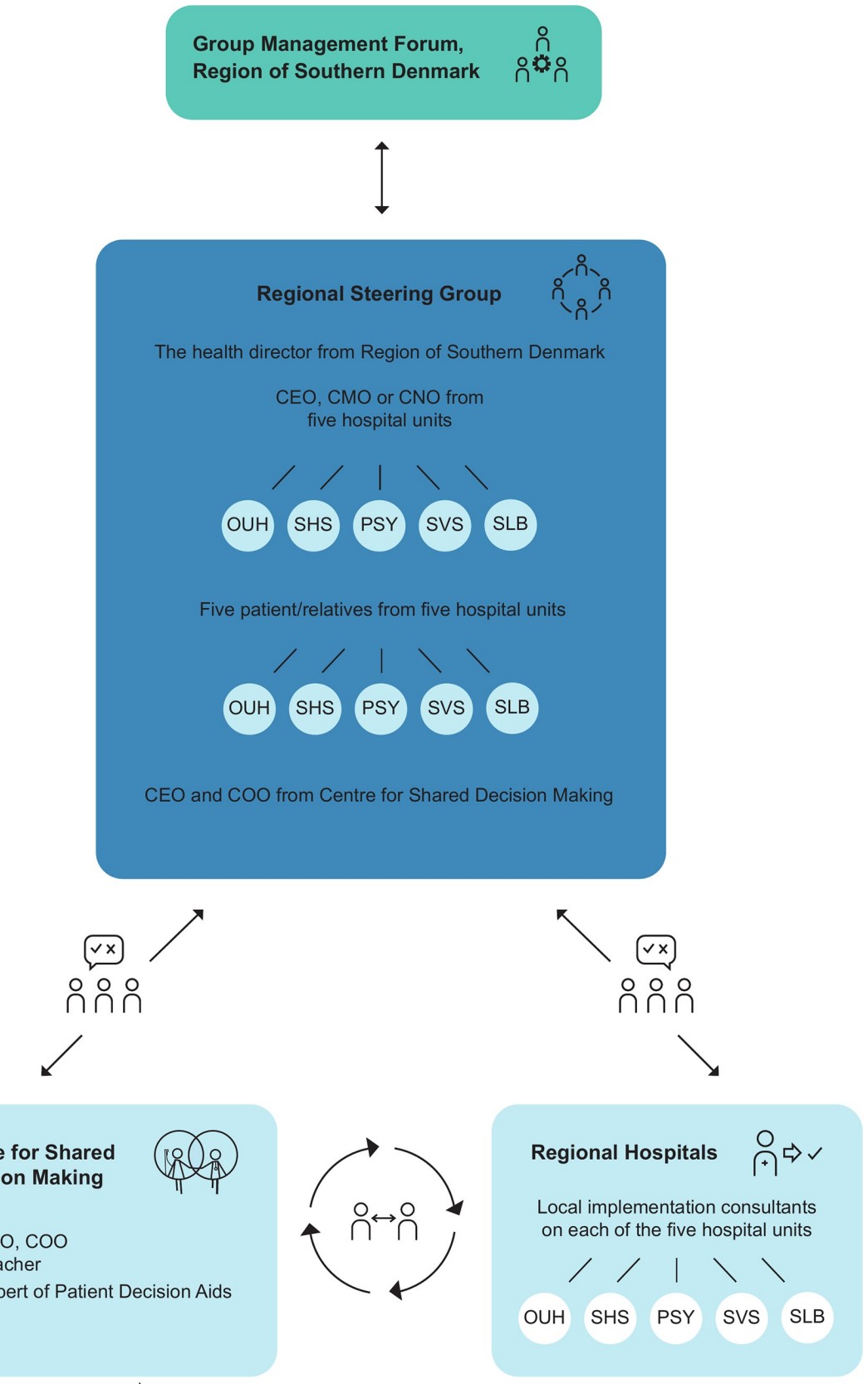

**Fig 2. Regional organization.**

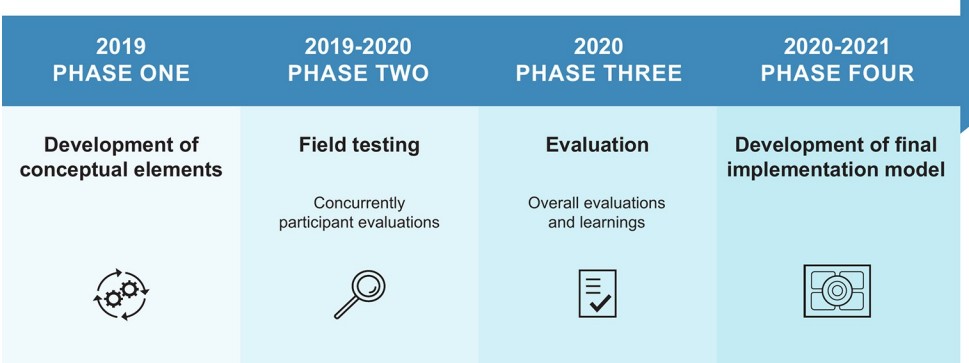

| 2019 PHASE ONE | 2019-2020 PHASE TWO | 2020 PHASE THREE | 2020-2021 PHASE FOUR |
|---|---|---|---|
| Development of conceptual elements | Field testing<br><br>Concurrently participant evaluations | Evaluation<br><br>Overall evaluations and learnings | Development of final implementation model |

**Fig 3. Timeline with activities.**

*Training of leaders.* Training of leaders at various levels of hospital departments aimed to provide knowledge and comprehend key elements of SDM and awareness of the important role of leadership during the SDM implementation process. A three-hour course, with a workshop format and a limit of 30 participants to ensure interactivity, was deemed feasible. It included evidence of SDM, overall goals of the implementation strategy, time for reflections of the relevance and applicability of SDM in own department and discussion on how to apply their specific roles as SDM leaders. The content of the training draws on general knowledge of leadership in improvement and project settings [28], but also specifically on the importance of leadership when implementing SDM [29].

*Training of teachers.* Training of teachers aimed to prepare clinicians to train own colleagues in SDM and in the use of PtDAs. At the same time, it provided every department with clinicians who possess special competencies and responsibilities within the department implementation. The training of teachers is built on the principle of training the trainers, which has been a recognized method at Lillebaelt Hospital when training clinicians' general communicative skills [30, 31]. The SDM course was developed as a two days course, with fourteen days between the course days. In this period, participants had homework, where they were asked to study their own practice within the perspective of SDM. The content of the course was mainly knowledge of SDM (day one1), training and simulation of patient/clinicians conversations using an adapted version of Glyn Elwyn's Three-Talk-Model [32] (day one and two), and preparing the trainers to develop their own teaching-class (day two). SDM as an approach to patients and relatives was highlighted as not only a way of communication, but as an approach–a concept that targets involvement of patients in medical decision making to a higher extent (day one and two). There was a limit of 16 participants on the course, to ensure individual profit of the course. After training the teachers, they were provided with different supporting materials to plan their teaching of colleagues.

*Training of clinicians.* The training of clinicians aimed to provide all clinicians with basic knowledge of SDM as a platform to further develop skills and behaviors in individual patient consultations. The new teachers were supported by the local implementation consultant in the process of planning local classes and taking on the responsibility as facilitator of courses. It was recommended, that a clinician course should make up at least a full day, but no hard rules of this were adopted.

*Patient Decision Aids (PtDAs).* Acknowledging that high-quality PtDAs [3] are strong facilitators of SDM, this conceptual element was integrated into the process. The PtDAs developed throughout the implementation phase, was based on an already thorough tested template, the

"Decision Helper template" [33]. The template is a generic template, where certain elements are mandatory and developed according to IPDAS criteria [34]. Other elements can be altered to fit the clinical context. The template has been tested through research studies in various clinical situations [21, 23–25]. PtDAs were developed multidisciplinary in working groups including clinicians, patients and/or relatives, and an expert from CFFB. To support the development process, health care professionals had access to an online platform hosting a template for systematic building of the PtDAs. The online platform offers step-by-step guidance to build and print individual Decision Helpers with the wanted clinical content, making it intuitive and easy to work with and providing healthcare professionals the opportunity to build and develop PtDAs tailored to their specific needs. Realizing that development of PtDAs is a complex venture, CFFB offered various degrees of support during the development and test phases, and were the experts ensuring that high quality PtDAs were developed. Working groups developing new PtDAs were guided through a systematic PtDA development process, with a literature review, workshops and alfa-and beta testing, ending with a final PtDA to fit the specific clinical situation. The development process was highly inspired by Coulters systematic development model [35].

*Systematic planning of the implementation process.* For both leaders, teachers and clinicians the question of HOW to practically implement SDM into routine clinical practice was substantial. A systematic approach of planning was obvious and manifested in different materials and support systems. A load-bearing element was an 'implementation starter kit', based on methods from System of Profound Knowledge and The South Danish Improvement Model. The implementation starter kit consisted of a one-year process plan with a fixed series of meetings with leaders and selected key persons from the department, standard agendas for these meetings, information to department managements, and PowerPoint slide-kits with information on SDM and information about the regional implementation task and goals. The process plan also included the other conceptual elements, i.e. Training of Leaders, Training of Teachers, Training of Clinicians, and development of a Decision Helper (optional). It was highlighted that the implementation starter kit was an example of how one <u>could</u> plan the local implementation, not how one <u>should</u> plan it. It gave room for local adjustments and alternative methods and feelings of ownership, and it furthermore provided learning for all involved.

*Support from Center for Shared Decision Making.* From the very beginning, leaders and staff members from CFFB played important roles in the processes described. Besides being responsible for the development of the conceptual elements, they took active part in the different activities, disseminated knowledge on SDM and implementation, initiated and supported a network between local implementation consultants, developed learning activities like an E-learning program and animated videos for use among professionals, and designed all evaluations of the process and collected results to develop the final implementation model. With an implementation effort of this size, several support systems are crucial. CFFB was responsible for integration of all elements and outcome measures in the existing regional and local workflows and IT systems, to secure quality and accessibility.

**Phase 2: Field testing.** Field-testing of the different elements was conducted during local implementation of SDM as part of quality improvement of routine clinical practice. Two to three first-mover departments at each of the five regional hospital units took part. Based on local decisions on which departments to include first, these pioneers covered a variety of departments including maternity wards, gynecology, oncology, kidney diseases, eating disorders, hematology and a team of clinical dietitians.

Each department carried out its prioritized and locally adapted activities during a period of around one year. They were continuously supported by the local SDM implementation consultants and individuals from CFFB.

During the field-testing phase, formal participant evaluations of the conceptual elements were given high priority to obtain first-hand evaluations from the clinicians participating in the first mover departments. It was sought to learn as much as possible during the project. Ongoing inspiring discussions were important for the understanding of feasibility and working mechanisms and led to important learnings during the implementation process.

**Phase 3: Evaluation of activities and overall implementation.** For evaluation of activities, three surveys were developed; i) Participants at the 'train the leaders course' were invited for a survey after one month covering the aims of the course ii) Participants of the teach-the-teachers course were asked to evaluate each course day separately and an overall evaluation, covering the overall perception of the course, the course content and general gain, and iii) one participant from each of the PtDAs groups evaluated the process on behalf of the whole group, including perceived relevance of the different elements of the process and satisfaction with the support from CFFB.

By the end of the first year of field-testing, a separate fourth questionnaire covered evaluation of the implementation process in itself as well as the implementation starter kit. Participants of the overall evaluation were leaders, key persons and clinicians in the first-mover departments. This evaluation was performed by CFFB. A questionnaire was developed where questions were answered on a 5-point Likert scale and included open questions about the overall implementation, respondents' specific role in the implementation process and the specific activities, which they had participated in.

With respect to improvement methodology thinking, where measures are used to accelerate improvements and tell if the changes made actually lead to improvement, it was important to create questions for the questionnaires to fit the specific element evaluated. The construction of all the questions were based on methods and findings from previous research and quality improvement projects [21, 36, 37], but were tailored to provide answers and useful evaluations about the specific element evaluated. Since it was a quality assessment study and not isolated research, the questionnaires were not rigorously tested. However, all questionnaires were tested several times in a PDSA-like process before use. Firstly in an internal process where four employees at CFFB reviewed the questions, adaptations were made, and then eight other employees reviewed the questionnaires. Afterward, external collaborators were involved in the review process, among them a survey specialist from the hospital's department of quality and a senior consultant from the National Center for Public and Private Innovation in Copenhagen [38]. All questionnaires was sent out electronically using SurveyXact (Ramboll Management Consulting, Aarhus, Denmark).

**Phase 4: Development of the final implementation model.** The results from the participant evaluations in the field-testing phase and the overall evaluation in phase three were themed within every conceptual element, to achieve a full view of all details of the concept, elucidated by different stakeholders and perspectives of the process. As overall responsible for the implementation process, the steering group committee took part in a three-hour virtual workshop, to further discuss the results and reach a consensus on specific elements and the need for local adaption. The virtual workshop consisted of a presentation of important evaluation data from every conceptual element, a "fish-bowl" discussion between two members of the steering committee followed by a group discussion to make the final decisions. The workshop was facilitated by CFFB. After the virtual workshop, the first draft of the model was developed. A few elements needed to be discussed further, which was done in a steering group meeting and resulted in minor adjustments of the implementation model before final approval.

**Analysis.** Since the main elements from this study are based on findings from surveys, statistical analysis was restricted to descriptive statistical methods. The analysis was supported by a survey expert from the hospital's Department of Quality, who is experienced in both

SurveyXact, descriptive statistics and data summaries. The survey expert used the analysis tools in SurveyXact to create descriptive reports of all questionnaires. Of the 5-point Likert scale questions, 4 and 5 were defined as 'top-scores'.

Comments and statements from the open-ended questions were sorted and themed into the four phases and were included in Fig 5.

**Ethical considerations.** All respondents of the surveys provided written consent in connection with answering the survey to the use of their response and their data for dissemination of knowledge about the evaluation and for further development of the implementation model.

According to the National Danish Consolidation Act on Research Ethics Review of Health Research Projects, Consolidation Act number 1083 of 15 September 2017 section 14 [39] approval of questionnaire surveys and medical database research projects to the system of research ethics committee system is only required if the project involves human biological material. Thus, the study was conducted without approval from The Committee on Health Research Ethics, as approval for this kind of study is not required according to Danish law.

## Results

### Development of conceptual elements

As a result of the initial development phase, the conceptual elements included in the field testing were: Separate courses targeting 1) training of leaders of departments, 2) clinicians who should take on the responsibility of teaching and training own colleagues, 3) clinicians who should learn about SDM and change their clinical practice, 4) development of PtDAs and, 5) elements and decisions related to the local implementation process including the implementation starter-kit with process plan, teaching materials, etc. Moreover, the level of support from CFFB was part of the field-testing.

### Training of leaders

Sixty-one department leaders participated in the leader training. All first-mover departments assigned leaders to the course. The majority (54%) were nurses, 33% physicians and the rest represented midwives, psychologists, therapists and other groups of leaders.

The evaluation surveys reached a response rate of 51%. Ninety-three and a half percent of the responders found the course relevant to some extent or a great deal. Seventy-one percent gave the course a top score (S1 Fig and S1 Table).

### Training of teachers

Eighty-four clinicians took part in the Teach-the-Teachers" course. Fifty-seven percent were nurses and 31% physicians. All first-mover departments assigned clinicians to participate in the course. Each first-mover department educated 1–4 teachers.

The evaluation survey had an overall response rate of 61%. S2 Table in the supplementary materials shows that there is a slight difference in the number of respondents from day one, day two, and the overall assessment. This is explained by sickness or other absences from day to day, or the incapability to evaluate on location. After the course, 96% felt prepared to plan and perform training of their colleagues. Ninety-three and a half percent gave the course a top score (S2 Fig and S2 Table).

### Training of clinicians

The 84 clinicians who had participated in the Teach-the-Teachers course, achieved to teach and train 547 clinicians during the first year. No systematic evaluation was performed of this element.

## Patient Decision Aids (PtDAs)

During the implementation process, the development of 19 Decision Helpers was initiated, involving 19 working groups with 37 members in total. Ten working groups had finalized their Decision Helper at the time of survey, nine were still in progress. Although COVID-19 led to restrictions for meetings and especially those with patients and relatives, eight of the 11 groups succeeded to conduct a user-involving workshop with participation of both patients, relatives and clinicians.

Six out of eight groups found it greatly relevant to involve patients and relatives in workshops when developing a Decision Helper, and seven out of eight groups would recommend doing a workshop when developing a Decision Helper (S3 Fig and S3 Table).

## Support from Center for Shared Decision

Eight out of ten responders from the regional steering committee and the local implementation consultants reported the support from CFFB to be useful in the local implementation to a high extent or very high extent. Eighty percent of clinicians found support from CFFB to be sufficient when developing a Decision Helper and/or in the implementation process (Fig 4 and S4 Table). Ninety-six % of participants in Teach-the-Teachers course reported that the trainer from CFFB was well prepared (S2 Fig and S2 Table).

**Overall evaluation.** An overall evaluation of the implementation process was conducted (Fig 4 and S4 Table). Activities from the field-testing phase and elements of the implementation starter kit were included in this overall evaluation. Eighty-seven members of the steering group committee, implementation consultants, and health care professionals from the implementation first-mover departments participated in the evaluation. The participants from the group of health care professionals had different roles in the implementation and were only asked questions relevant to their role. This explains the various numbers of respondents to the questions in S2 Fig and S2 Table. Only questions with a number above 20 respondents are included in the table.

Sixty-one percent had been using a process plan from quite a bit to a lot, 89.5% reported that pre-meeting with department leaders were useful and 74% agreed that a series of fixed meetings were relevant. Majority of responders thought that the elements "teach-the-teachers", Training of leaders, training of clinicians and development of a Decision Helper were useful from quite a bit to a great deal.

**Learnings from the implementation process.** As illustrated in Fig 5, the field-testing phase was characterized by a high level of learning, adjusting, and sharing experiences. Elements were adjusted, both locally and on a regional level. Adjustment at the regional level, i.e. across local entities (departments/hospital units) always included reflections and decisions of the regional steering committee. Local adjustments were made to customize the process to the specific local workflows and culture, but with the overall perspectives in mind, to ensure similarity in the process across the Region of Southern Denmark.

**Development of the final implementation model.** During the final revision of the model. it was agreed to keep the three themes i)Training of Leaders, ii) Training of Teachers and Clinicians, iii) Decision Helper, and to add a fourth theme entitled 'Process'. This new theme emerged since the overall evaluation pointed at several elements, which all were a part of the systematic implementation process, and which were evaluated as very useful (Fig 4). The first three elements included were all conceptual elements from the beginning, however, the fourth theme "Process" and the details of this theme, stood out as necessary foundation of a successful implementation of SDM.

Next, it became clear that there were three different levels of elements within each theme; shared elements, meaning that all hospitals should include these elements in their

# Overall evaluation

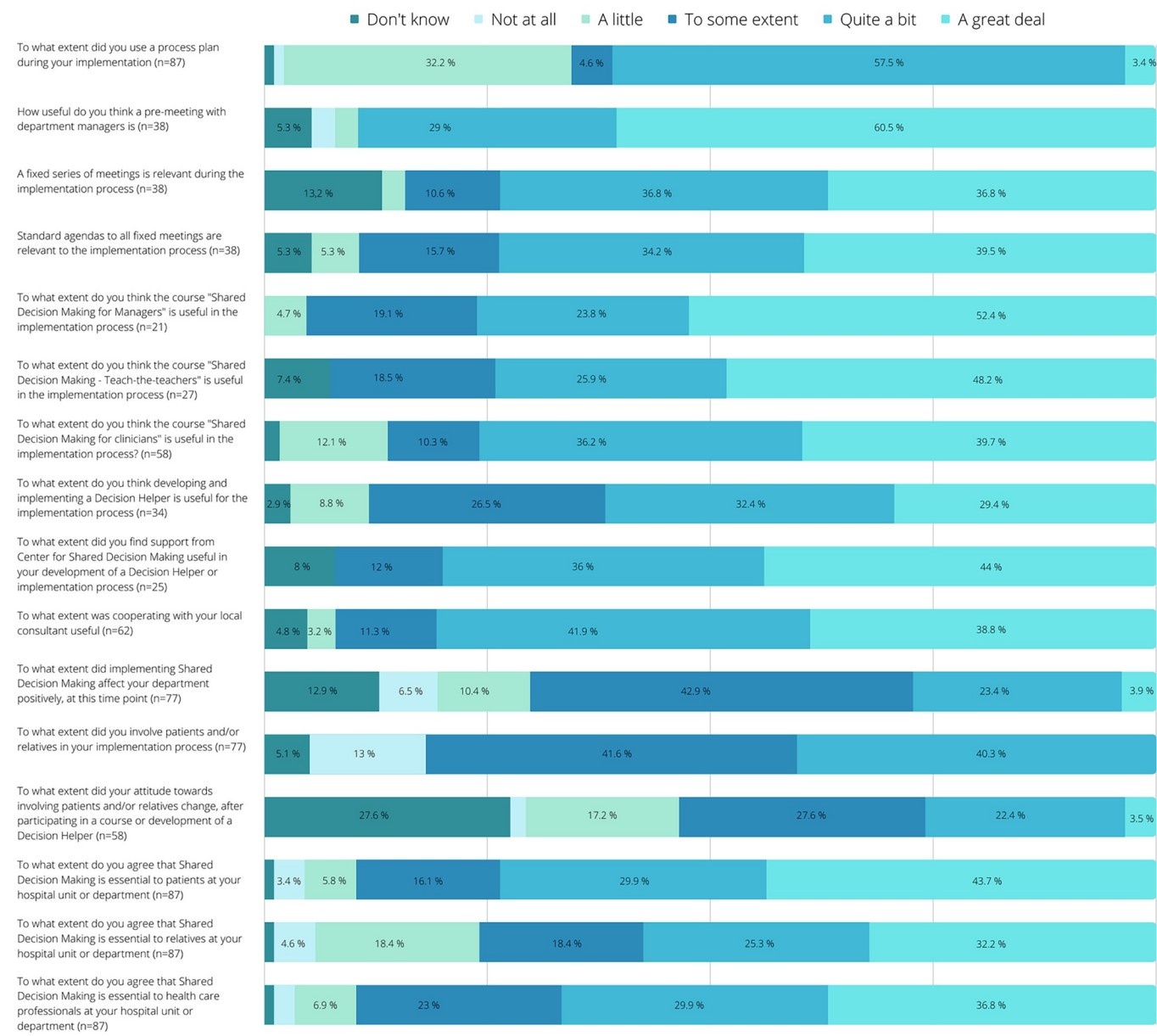

**Fig 4. Overall evaluation of the implementation process.**

implementation of SDM, recommendations to be aware of during implementation if possible, and at last elements that could be locally adjusted. The final model therefore ended with the three levels of elements, 1) shared elements, 2) recommendations, and 3) local adaption.

After a last revision, the final model was described in detail and illustrated graphically (Fig 6).

Details of Training of Leaders, Training of Teachers and Clinicians, Decision Helper and the theme Process are shown in S4 Fig. A full overview of the model in a table edition can be seen in S5 Fig.

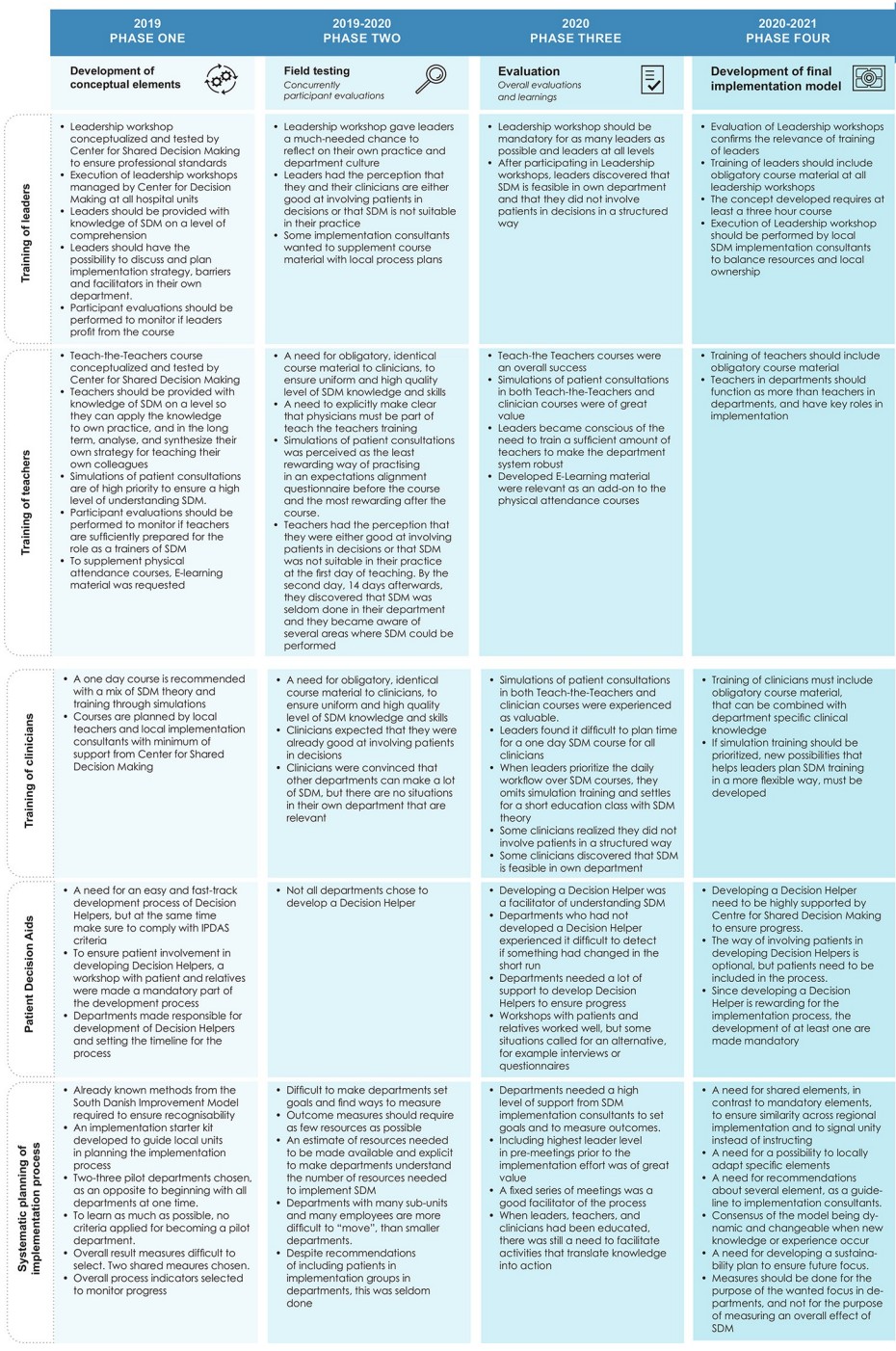

**Fig 5.** A and B Learnings from the implementation process.

## Discussion

The SDM:HOSP model of implementation of SDM presented here is the primary outcome of the SDM implementation process that has taken place in the Region of Southern Denmark during the last 2½ years. The model emerged from fieldwork highly inspired by the literature, theoretical frameworks and systematic processes with a high level of user involvement.

**SDM:HOSP**

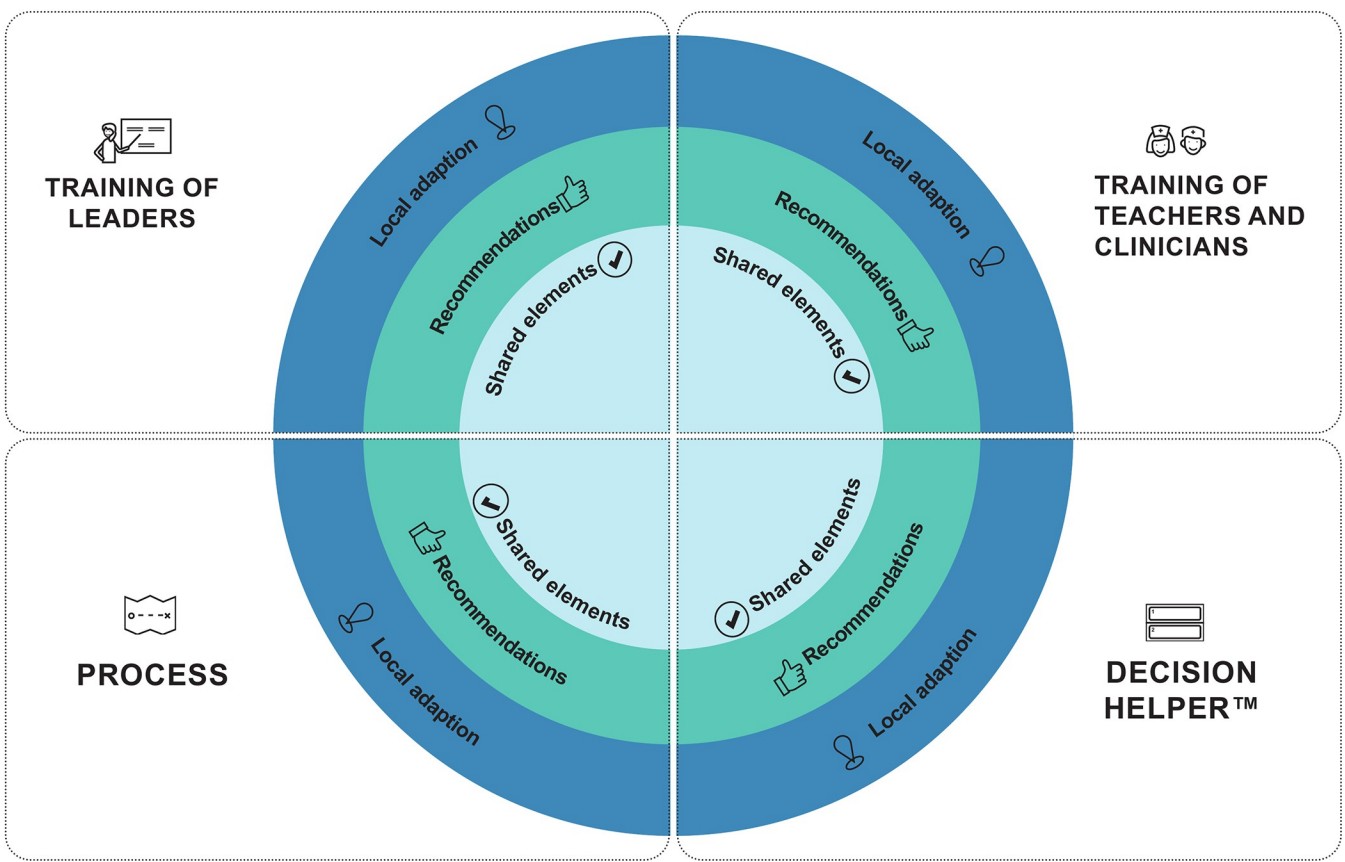

**Fig 6. The final model SDM:HOSP.**

When working with the implementation of initiatives that require change of culture and attitudes as well as sustainable change management, the challenge is often to prove that the effort has led to improvements. SDM implementation does not deal with the implementation of a new diagnostic test or a new pharmaceutical drug where clear and well-known improvement measures such as improved sensitivity, positive predictive value, or altered number of events exist. Implementation is the result of complex interventions and initiatives. Alternative outcomes may cover processes rather than results, and result outcome measures are often complex, reflecting various factors than the specific effort one is trying to evaluate. Patient experienced level of SDM is an example. In the current implementation initiative, it was therefore prioritized to systematically collect evaluations from all kinds of course participants (leaders, teachers, clinicians), steering group members and the local implementation consultants. In addition, several other process measures were collected (e.g. number of trained leaders, teachers, clinicians and developed PtDAs) as well as some performance goals, which, however, are outside the scope of the present paper as they only exist at a specific department level.

This Danish initiative on SDM implementation is not unique; other organizations have also worked with large-scale SDM implementation like us, for example the national effort in Norway where "Samvalg" is the Norwegian term for "shared decision making" [40], or the "Share to Care" initiative at the University Hospital Schleswig-Holstein, Campus Kiel in Germany

[41] who recently published a German-language systematic methodology report [42] on their website. Furthermore, there has been a major effort in the UK where the Health Foundation in the UK commissioned the MAGIC (Making Good Decisions in Collaboration) program to design, test, and identify the best ways to embed SDM into routine primary and secondary care [9]. This initiative likewise used quality improvement methods.

A common feature of these efforts is they all point out how important it is to increase understanding of what SDM entails and how it differs from current clinical practice. They all include a common denominator that deals with training of clinicians, often in workshops based on role-play-based training or as feedback on real-world video recording of the participating clinician's consultations with real patients from their practice. These are also key elements of our clinician training program. Another common denominator is the development and implementation of PtDAs. It is well known that both policymakers and many clinicians believe that a decision aid will itself enable SDM and believe that a tool to provide to their patients is essential. A key learning point from the MAGIC program was that "skills trump tools, and attitudes trump skills." These are findings that we also recognize from our implementation. One of the four themes of our final model was therefore 'Training of leaders'; they are key opinion leaders that shape the culture and attitudes in an organization. We cannot emphasize enough how important this is. The leaders will need to both 'talk the talk' and 'walk the walk' putting action behind their words to secure that SDM can be effectively implemented in a large organization like a hospital.

There is no doubt that PtDAs are often a great facilitator of SDM implementation processes in general. As part of this, clinicians who participate in the development often gain much greater interest and insight into what SDM means and how it e.g. can be practiced with the use of a patient decision aid. When the PtDAs are based on an already thorough tested template, like in this case, the benefits of including the clinicians with highly specialized knowledge and insight into the decision in question is evident. It creates ownership and helps to change a culture, and the discussions caused by the PtDAs among the clinicians who help to develop them are often very fruitful during the development phase. As part of a process updating evidence for the International Patient Decision Aid Standards Collaboration (IPDAS) Checklist, a rapid realist review was conducted in 2020 and described recommendations that point in the same direction as ours including training clinical teams, co-production of PtDAs, measurements, senior-level buy-in and in preparing patients to participate in SDM [13].

We would therefore strongly recommend involving–and supporting—the clinicians who are going to use PtDAs in the development process of those and not to outsource it to external developers. In addition, at the same time be aware that PtDAs cannot stand alone.

The latter, patient activation in the format of preparing patients for their individual participation in SDM, has not been a targeted systematic effort in our implementation process. However, we have consequently involved patients in the designing of the PtDAs, in raising awareness and disseminating the concept of SDM based on patient campaigns in Vejle Hospital in Denmark [43], and in the production of an animation video targeted at patients describing what SDM is and why it is important [44]. We fully recognize that education and activities targeting how to prepare patients to engage in SDM are essential, as this is a serious practical barrier for SDM implementation.

During this SDM initiative it became clear how organizational and cultural differences including the various hierarchical models of communication and decision-making at department and hospital level constituted contextual factors of importance to the ongoing processes and local adaptation. Even though The Region of Southern Denmark is a relatively small area with approximately alike demographics and living conditions for the citizens and an executive board that applies the "rules" of health care, the five hospital units applied very differently.

This called for a very flexible and adaptable support from CFFB and the need for a flexible model for future use. While some hospital units wanted to mostly decide as much as possible on their own, others wanted a more strict and united framework. To secure feasibility across entities and similarity on the main elements of implementation, we introduced the different levels 'shared', 'recommended' and 'adapted'

If the four themes conceptualized by the three levels are applied to hospital settings when implementing SDM, one would hypothesize that the implementation includes the most important elements and working mechanisms and address the most crucial barriers, thereby improving SDM in clinical practice. While there is plenty of knowledge about the need for training clinicians and developing PtDA's when implementing SDM, the two other themes in SDM: HOSP are, in our opinion, equally important. Encouraging and motivating leadership in a complex intervention like SDM implementation is crucial. The fourth theme of "Process" includes elements of improvement methodology, which forms a concrete and clear foundation when planning and executing SDM implementation.

We, therefore, suggest that our adaptive model (SDM:HOSP) will be of value when implementing SDM across departments within one hospital and/ or when implementing SDM across hospitals sites and can make a feasible framework for SDM implementation, both inside the borders of Denmark and possibly also internationally.

## Strengths and limitations

It is a strength that the model was developed during a rigorous process taking outset in routine clinical practice across different hospital settings. Second, it is a strength that all steps of the process included a high level of user involvement and systematic evaluation including different perspectives and both qualitative and quantitative methods. Third, the model was developed across hospital units including both mental and somatic units, clinical departments representative of different clinical problems, medical specialties and both nurses and medical doctors. A fourth strength of this work was the possibility to work in a big "living lab", with different organizations and cultures. This means a high level of learning of what is needed in different contexts when implementing SDM, and a realization that activities both need to be shared, but not so tight that activities cannot be adjusted according to the local context. There are several limitations to this work. First, the design of the intervention is pragmatic to be feasible to real-life practice, and this means that there have been several adjustments, both regionally and locally, during the intervention. Secondly, it is a limitation, that the usefulness and effectiveness of the final model have not yet been tested.

## Perspectives

As always when implementing SDM, it is a challenge to provide measures to detect if the effort results in an actual improvement and change. This calls for an innovative approach to support leaders and clinicians, for example by developing a tool for setting goals of implementation, choosing implementation activities and measuring improvement and progress of implementation. This would be a practical and usable add-on to SDM:HOSP, where SDM:HOSP would function as the overall framework and a supportive tool for the above purposes would make the implementation of SDM a much more practical oriented process. Development of such a supportive tool is an ongoing project in CFFB,

## Conclusion

To our knowledge, this is the first feasible and acceptable generic model for implementation of SDM across hospitals and departments that accounts for different organizations and cultures.

Although the details of the model are adapted to a Danish, regional context, the overall design with the four themes and three levels can easily be adapted to other organizations, and the generic touch of details in the themes and levels can be adjusted to fit the specific organization and culture. The results from the ongoing and overall evaluation suggest promising avenues for future work in further testing and research of the usability of the model.

## Supporting information

**S1 File. Checklist squire guidelines.**
(DOCX)

**S1 Fig. Training of leaders.**
(TIFF)

**S2 Fig. Training of teachers.**
(TIFF)

**S3 Fig. Development of decision helpers.**
(TIFF)

**S4 Fig.** A-D. Elaborated themes within SDM:HOSP.
(TIFF)

**S5 Fig. Overview of SDM:HOSP, Table edition.**
(TIFF)

**S1 Table. Training of leaders.**
(DOCX)

**S2 Table. Training of teachers.**
(DOCX)

**S3 Table. Development of decision helpers.**
(DOCX)

**S4 Table. Overall evaluation of the implementation process.**
(DOCX)

## Acknowledgments

We sincerely thank the implementation consultants and employees at Center for Shared Decision Making for their great contribution to all of the elements described in this quality improvement project. We extend our appreciation to the patient-relatives from the steering committee, who provided feedback and questions to adjust accordingly to their preferences. We also acknowledge the first mover departments in the Region of Southern Denmark, who all willingly participated in the project and made it possible to get experience and learnings to develop the final model of implementation.

## Author Contributions

**Conceptualization:** Karina Dahl Steffensen, Kurt Espersen, Susanne Lauth, Peter Fosgrau, Anders Meinert Pedersen, Peter Sigerseth Groen, Christian Sauvr, Karina Olling.

**Data curation:** Karina Dahl Steffensen, Karina Olling.

**Formal analysis:** Karina Dahl Steffensen, Karina Olling.

**Investigation:** Karina Dahl Steffensen, Karina Olling.

**Methodology:** Karina Dahl Steffensen, Karina Olling.

**Project administration:** Karina Olling.

**Resources:** Karina Dahl Steffensen, Kurt Espersen, Susanne Lauth, Peter Fosgrau, Anders Meinert Pedersen, Peter Sigerseth Groen, Christian Sauvr.

**Supervision:** Karina Dahl Steffensen, Karina Olling.

**Validation:** Karina Dahl Steffensen, Karina Olling.

**Visualization:** Karina Dahl Steffensen, Kurt Espersen, Susanne Lauth, Peter Fosgrau, Anders Meinert Pedersen, Peter Sigerseth Groen, Christian Sauvr, Karina Olling.

**Writing – original draft:** Karina Dahl Steffensen, Dorte Gilså Hansen, Karina Olling.

**Writing – review & editing:** Karina Dahl Steffensen, Dorte Gilså Hansen, Kurt Espersen, Susanne Lauth, Peter Fosgrau, Anders Meinert Pedersen, Peter Sigerseth Groen, Christian Sauvr, Karina Olling.

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
