## [Decision Letter · Decision Letter 0]

30 Aug 2022

PONE-D-22-19504“SDM:HOSP”- a generic model for hospital-based implementation of Shared Decision MakingPLOS ONE

Dear Dr. Olling,

Thank you for submitting your manuscript to PLOS ONE. After careful consideration, we feel that it has merit but does not fully meet PLOS ONE’s publication criteria as it currently stands. Therefore, we invite you to submit a revised version of the manuscript that addresses the points raised during the review process.

Both reviewers are positive about the manuscript. One of them has, however, some minor remarks that I kindly ask you to consider before I can make a final decision about publication. 

We look forward to receiving your revised manuscript.

Kind regards,

Sara Rubinelli

Academic Editor

PLOS ONE

Journal Requirements:

3. Please clarify whether participants gave their consent for their data to be used in this study.

"No funding for this project was obtained except for institutional funding for the salary of the involved participants."

Reviewers' comments:

Reviewer's Responses to Questions

**Comments to the Author**

1. Is the manuscript technically sound, and do the data support the conclusions?

Reviewer #1: Partly

Reviewer #2: Yes

2. Has the statistical analysis been performed appropriately and rigorously? 

Reviewer #1: No

Reviewer #2: N/A

3. Have the authors made all data underlying the findings in their manuscript fully available?

Reviewer #1: Yes

Reviewer #2: Yes

4. Is the manuscript presented in an intelligible fashion and written in standard English?

Reviewer #1: Yes

Reviewer #2: Yes

5. Review Comments to the Author

Reviewer #1: Dear Authors, I am happy to see an implementation science paper came to my review table almost fully equipped. WhileI do not have qualms on the development procedures of the study, I do have some comments for improvements for the manuscript as they are some missing information or better way of presenting the information:

1. There is missing information on all four the questionnaires used.

a. What are the basis of these questionnaire?

b. Were they self-created or adapted?

c. If adapted, please provide the necessary reference?

d. If modifications were made in the adaptation, please detail the procedures and pilot outcomes.

e. If its self-created, this will lead to a whole new string of procedures that is quite lengthy but needed to be explained in this manuscript or referenced to possibly another development and validation paper.

2. The analysis section is too brief. Kindly detail the analysis procedure if they are descriptive or inferential in nature and justify why they are chosen.

3. Data presentation in supporting information section:

a. While the presentation of the questionnaire is highly recommended, the data could be presented better in the form of graphs or charts that is beneficial in terms of visual representation. Presentation of plain numbers in questionnaire layout is too crowded and confusing.

b. The presentation of the results in the manuscript is also inconsistent to the presentation in the supporting information. While the supporting information provided the raw numbers, the manuscript results were presented in percentage and as a reader I find this confusing and inconsistent. Please revise.

4. In the results of the Development of the final implementation model (line 357-358) and in the Discussion section (sporadically mentioned but a longer mention in line

Reviewer Recommendation and Comments for Manuscript Number PONE-D-22-21297

439-441) was the the introduction of the fourth theme -process. How did this "new theme" came around as there are no data to support the creation of this theme or the procedural generation of it. Please define this "missing link"in the manuscript.

Reviewer #2: This is a very good study. Shared Decision Making between healthcare professionals and patients within the hospital setting is something that should be applicable to almost all countries. This is very important. This study is a first, or at least among the first of its kind, and it is needed by many countries. Well done to the authors for coming up with such an important study.

6. PLOS authors have the option to publish the peer review history of their article (what does this mean?). If published, this will include your full peer review and any attached files.

Reviewer #1: **Yes: **Mohammad Zabri Johari

Reviewer #2: No

---

## [Author Response · Author response to Decision Letter 0]

17 Oct 2022

There is missing information on all four the questionnaires used.

a. What are the basis of these questionnaire?

b. Were they self-created or adapted?

c. If adapted, please provide the necessary reference?

d. If modifications were made in the adaptation, please detail the procedures and pilot outcomes.

e. If its self-created, this will lead to a whole new string of procedures that is quite lengthy but needed to be explained in this manuscript or referenced to possibly another development and validation paper.

Thank you for this important comment. We totally agree and completely recognize that this is important but missing information.

We have included a paragraph in the methods section under the sub-paragraph “Phase 3: Evaluation of activities and overall implementation (page 12). This is in order to more clearly show the basis of the questionnaires and the process of developing these. We have also included additional references (38 and 39) to show examples of what have inspired us in developing the questionnaires. See "Response for Reviewers" for revised text

The analysis section is too brief. Kindly detail the analysis procedure if they are descriptive or inferential in nature and justify why they are chosen.

Thank you for this comment. 

We have now elaborated on the analysis section. Please see "Response for Reviewers" for revised text

3. Data presentation in supporting information section:

a. While the presentation of the questionnaire is highly recommended, the data could be presented better in the form of graphs or charts that is beneficial in terms of visual representation. Presentation of plain numbers in questionnaire layout is too crowded and confusing.

b. The presentation of the results in the manuscript is also inconsistent to the presentation in the supporting information. While the supporting information provided the raw numbers, the manuscript results were presented in percentage and as a reader I find this confusing and inconsistent. Please revise.

This is a good point as well – thank you for that.

a. We have now made bar charts of all data in the tables as following; 

• Data from “Training of Leaders” are presented in a bar chart in S2 Fig as well in S3 Table

• Data from “Teach-the-Teachers” are presented in a bar chart in S4 Fig and S5 Table

• Data from “Development of Decision Helpers” are presented in S6 Fig and S7 Table

• Data from the overall evaluation of the implementation process are presented in Figure 4, which are now included in the main manuscript. Data in a table format from the overall evaluation are also included as supplementary material in S8 Table

b. The new bar charts are presented with percentages, hence this now creates better consistency with the text in the manuscript.

4. In the results of the Development of the final implementation model (line 357-358) and in the Discussion section (sporadically mentioned but a longer mention in line 439-441 was the the introduction of the fourth theme -process. How did this "new theme" came around as there are no data to support the creation of this theme or the procedural generation of it. Please define this "missing link"in the manuscript.

Thank you again for a really good and useful comment – we understand this can create confusion, and needs to be elaborated.

We have included additional text in the paragraph “Results” in the sub-paragraph “Development of the final implementation model". We have also elaborated on the subject in the Discussion paragraph. Please see "Response for Reviewers" for revised text.

This is a very good study. Shared Decision Making between healthcare professionals and patients within the hospital setting is something that should be applicable to almost all countries. This is very important. This study is a first, or at least among the first of its kind, and it is needed by many countries. Well done to the authors for coming up with such an important study.

Thank you very much for this very kind and appreciative comment. We hope you find the changes we made equally relevant and good.

---

## [Decision Letter · Decision Letter 1]

4 Jan 2023

“SDM:HOSP”- a generic model for hospital-based implementation of Shared Decision Making

PONE-D-22-19504R1

Dear Dr. Olling,

We’re pleased to inform you that your manuscript has been judged scientifically suitable for publication and will be formally accepted for publication once it meets all outstanding technical requirements.

Kind regards,

Sara Rubinelli

Academic Editor

PLOS ONE

Additional Editor Comments (optional):

Reviewers' comments:

Reviewer's Responses to Questions

**Comments to the Author**

1. If the authors have adequately addressed your comments raised in a previous round of review and you feel that this manuscript is now acceptable for publication, you may indicate that here to bypass the “Comments to the Author” section, enter your conflict of interest statement in the “Confidential to Editor” section, and submit your "Accept" recommendation.

Reviewer #1: All comments have been addressed

Reviewer #2: All comments have been addressed

2. Is the manuscript technically sound, and do the data support the conclusions?

Reviewer #1: Yes

Reviewer #2: Yes

3. Has the statistical analysis been performed appropriately and rigorously? 

Reviewer #1: Yes

Reviewer #2: N/A

4. Have the authors made all data underlying the findings in their manuscript fully available?

Reviewer #1: Yes

Reviewer #2: Yes

5. Is the manuscript presented in an intelligible fashion and written in standard English?

Reviewer #1: Yes

Reviewer #2: Yes

6. Review Comments to the Author

Reviewer #1: I am happy with all the revisions made and have no further comments to add. I support the publication of this manuscript as it meets the requirements based on the recent revision.

Reviewer #2: The authors have adequately addressed comments in a previous round of review. As such, the manuscript is acceptable for publication.

7. PLOS authors have the option to publish the peer review history of their article (what does this mean?). If published, this will include your full peer review and any attached files.

Reviewer #1: **Yes: **Dr Mohammad Zabri Johari

Reviewer #2: No

---

## [Editor Report · Acceptance letter]

6 Jan 2023

PONE-D-22-19504R1 

“SDM:HOSP”- a generic model for hospital-based implementation of Shared Decision Making 

Dear Dr. Olling:

I'm pleased to inform you that your manuscript has been deemed suitable for publication in PLOS ONE. Congratulations! Your manuscript is now with our production department. 

Kind regards, 

on behalf of

Dr. Sara Rubinelli 

Academic Editor

PLOS ONE